# On the Explanatory Power of Decision Trees

## Abstract

Decision trees have long been recognized as models of choice in sensitive applications where interpretability is of paramount importance. In this paper, we examine the computational ability of Boolean decision trees in deriving, minimizing, and counting sufficient reasons and contrastive explanations. We prove that the set of all sufficient reasons of minimal size for an instance given a decision tree can be exponentially larger than the size of the input (the instance and the decision tree). Therefore, generating the full set of sufficient reasons can be out of reach. In addition, computing a single sufficient reason does not prove enough in general; indeed, two sufficient reasons for the same instance may differ on many features. To deal with this issue and generate synthetic views of the set of all sufficient reasons, we introduce the notions of relevant features and of necessary features that characterize the (possibly negated) features appearing in at least one or in every sufficient reason, and we show that they can be computed in polynomial time. We also introduce the notion of explanatory importance, that indicates how frequent each (possibly negated) feature is in the set of all sufficient reasons. We show how the explanatory importance of a feature and the number of sufficient reasons can be obtained via a model counting operation, which turns out to be practical in many cases. We also explain how to enumerate sufficient reasons of minimal size. We finally show that, unlike sufficient reasons, the set of all contrastive explanations for an instance given a decision tree can be derived, minimized and counted in polynomial time.

## 1 Introduction

In essence, explaining a decision to a person is to give the details or *reasons* that help a person (the explainee) understand why the decision has been made. This is a significant issue especially when decisions are made by Machine Learning (ML) models, such as random forests, Markov networks, support vector machines, and deep neural networks. Actually, with the growing number of applications that rely on ML techniques, researches on eXplainable AI (XAI) have become increasingly important, by providing efficient methods for interpreting ML models, and explaining their decisions (see for instance [10, 11, 12, 13, 16, 19, 22, 23, 24, 28, 30]).

When dealing with Boolean classifiers, which is what we do in this paper, two decisions are possible, only: 1 for the instances classified as positive instances, and 0 for the remaining ones (the negative instances). Whatever the way $x$ has been classified, an explainee may seek for explanations from two distinct types [23]. On the one hand, abductive explanations for $x$ are intended to explain why $x$ has been classified in the way it has been classified by the ML model (thus, addressing the "Why?" question). On the other hand, the purpose of contrastive (also known as counterfactual) explanations for $x$ is to explain why $x$ has not been classified by the ML model as the explainee expected it (thus, addressing the "Why not?" question). In both cases, explanations that are as simple as possible are preferred (where simplicity is modeled as irredundancy, or even as size minimality).

Submitted to 35th Conference on Neural Information Processing Systems (NeurIPS 2021). Do not distribute.

Although there is no formal notion of *interpretability* [21], for classification problems, *decision trees* [3, 26] are arguably among the most interpretable ML models. Because of their interpretability, decision trees are often considered as target models for distilling a black-box model into a comprehensible one [4, 10]. Furthermore, decision trees are often the components of choice for building (less interpretable, but potentially more accurate) ensemble classifiers, such as random forests [2] and gradient boosted decision trees [5].

The interpretability of decision trees is endowed with two key characteristics. On the one hand, decision trees are *transparent*: each node in a decision tree has some meaning, and the principles used for generating all nodes can be explained. On the other hand, decision trees are *locally explainable*: by construction of a decision tree $T$, any input instance $\boldsymbol{x}$ is mapped to a unique root-to-leaf path that yields to a decision label. The subset of (positive and negative) features $t_{\boldsymbol{x}}^T$ occurring in the path used to find the right label 1 or 0 for $\boldsymbol{x}$ in the decision tree $T$ can be viewed as a "direct reason" for classifying $\boldsymbol{x}$ as a positive instance or as a negative instance. $t_{\boldsymbol{x}}^T$ is an abductive explanation for $\boldsymbol{x}$ given $T$, which explains why $\boldsymbol{x}$ has been classified by $T$ as it has been classified. Indeed, every instance $\boldsymbol{x}'$ that coincides with $\boldsymbol{x}$ on $t_{\boldsymbol{x}}^T$ is classified by $T$ in the same way as $\boldsymbol{x}$. However, such "direct reasons" can contain arbitrarily many redundant features [17]. This motivates to take account for other types of abductive explanations in the case of decision trees, namely, sufficient reasons [7] (also known as prime implicant explanations [29]), that are irredundant abductive explanations, and minimal sufficient reasons (i.e., those sufficient reasons of minimal size).

In this paper, we examine the computational ability of Boolean decision trees in deriving, minimizing and counting sufficient reasons and contrastive explanations. We prove that the set of all sufficient reasons of minimal size for an instance given a decision tree can be exponentially larger than the size of the input. When this is the case, generating the full set of sufficient reasons (i.e., the complete reason for the instance [7]) is typically out of reach. In addition, computing a single sufficient reason does not prove enough in general; indeed; two sufficient reasons for the same instance may differ on many features. To deal with this issue and generate synthetic views of the set of all sufficient reasons, we introduce the notions of relevant features and of necessary features that characterize the (possibly negated) features appearing in at least one or in every sufficient reason, and we show that they can be computed in polynomial time. We also introduce the notion of explanatory importance, that indicates how frequent each (possibly negated) feature is in the set of all sufficient reasons. Though deriving the explanatory importance of a feature in the set of sufficient reasons and determining the cardinality of this set are two computationally demanding tasks, we show how they can be achieved thanks to model counting operation, which turns out to be practical in many cases. We also explain how to enumerate sufficient reasons of minimal size, which is a way to count them when they are not too numerous. We finally show that, from a computational standpoint, contrastive explanations highly depart from sufficient reasons. Indeed, the set of all contrastive explanations for an instance given a decision tree can be computed in polynomial time. As a consequence, such explanations can also be minimized and counted in polynomial time.

The rest of the paper is organized as follows. Preliminaries about decision trees, abductive reasons, and contrastive explanations are given in Section 2. The computation of all sufficient reasons is considered in Section 3. Necessary and relevant features are presented in this section, as well as the approach for assessing the explanatory importance of a feature and for counting the number of sufficient reasons. We also explain there how minimal sufficient reasons can be enumerated. An algorithm for computing all the contrastive explanations for the instance given the decision tree is presented in Section 4. Experimental results are reported in Section 5. Finally, Section 6 concludes the paper. All the proofs and additional empirical results are reported as a supplementary material.

## 2 Decision Trees, Abductive and Contrastive Explanations

For an integer $n$, let $[n]$ be the set $\{1, \cdots, n\}$. By $\mathcal{F}_n$ we denote the class of all Boolean functions from $\{0, 1\}^n$ to $\{0, 1\}$, and we use $X_n = \{x_1, \cdots, x_n\}$ to denote the set of input Boolean variables, corresponding to the features under consideration. Any assignment $\boldsymbol{x} \in \{0, 1\}^n$ is called an *instance*. If $f(\boldsymbol{x}) = 1$ for some $f \in \mathcal{F}_n$, then $\boldsymbol{x}$ is called a *model* of $f$. $\boldsymbol{x}$ is a *positive instance* when $f(\boldsymbol{x}) = 1$ and a *negative instance* when $f(\boldsymbol{x}) = 0$.

We refer to $f$ as a *propositional formula* when it is described using the Boolean connectives $\wedge$ (conjunction), $\vee$ (disjunction) and $\neg$ (negation), together with the Boolean constants 1 (true) and 0

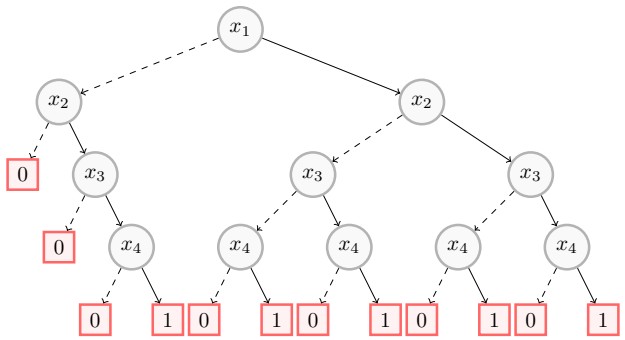

Figure 1: A decision tree $T$ for recognizing *Cattleya* orchids. The left (resp. right) child of any decision node labelled by $x_i$ corresponds to the assignment of $x_i$ to 0 (resp. 1).

(false). As usual, a *literal* $\ell$ is a variable $x_i$ (a positive literal) or its negation $\neg x_i$, also denoted $\overline{x}_i$ (a negative literal). A positive literal $x_i$ is associated with a positive feature (i.e., $x_i$ is set to 1), while a negative literal $\overline{x}_i$ is associated with a negative feature (i.e., $x_i$ is set to 0). A *term* (or *monomial*) $t$ is a conjunction of literals, and a *clause* $c$ is a disjunction of literals. A DNF *formula* is a disjunction of terms and a CNF *formula* is a conjunction of clauses. The set of variables occurring in a formula $f$ is denoted $Var(f)$. A formula $f$ is *consistent* if and only if it has a model. A CNF formula is *monotone* whenever every occurrence of a literal in the formula has the same polarity (i.e., if a literal occurs positively (resp. negatively) in the formula, then it does not have any negative (resp. positive) occurrence in the formula). A formula $f_1$ *implies* a formula $f_2$, noted $f_1 \models f_2$, if and only if every model of $f_1$ is a model of $f_2$. Two formulae $f_1$ and $f_2$ are *equivalent*, noted $f_1 \equiv f_2$ whenever they have the same models. The *conditioning* of a formula $f$ by a literal $\ell$, denoted $f \mid \ell$, is the formula obtained from $f$ by replacing each occurrence of $x_i$ with 1 (resp. 0) and each occurrence of $\overline{x}_i$ with 0 (resp. 1) if $\ell = x_i$ (resp. $\ell = \overline{x}_i$).

In what follows, we shall often treat assignments as terms, and terms and clauses as sets of literals. Given an assignment $\boldsymbol{z} \in \{0, 1\}^n$, the corresponding term is defined as

$$t_{\boldsymbol{z}} = \bigwedge_{i=1}^{n} x_i^{z_i} \text{ where } x_i^0 = \overline{x}_i \text{ and } x_i^1 = x_i$$

A term $t$ *covers* an assignment $\boldsymbol{z}$ if $t \subseteq t_{\boldsymbol{z}}$. An *implicant* of a Boolean function $f$ is a term that implies $f$. A *prime implicant* of $f$ is an implicant $t$ of $f$ such that no proper subset of $t$ is an implicant of $f$. Dually, an *implicate* of a Boolean function $f$ is a clause that is implied by $f$, and a *prime implicate* of $f$ is an implicate $c$ of $f$ such that no proper subset of $c$ is an implicate of $f$.

With these basic notions in hand, we shall focus on the following representation class of Boolean functions:

**Definition 1** (Decision Tree). *A (Boolean) decision tree is a binary tree $T$, each of whose internal nodes is labeled with one of $n$ input Boolean variables, and whose leaves are labeled $0$ or $1$. Every variable is assumed (without loss of generality) to appear at most once on any root-to-leaf path (read-once property). The value $T(\boldsymbol{x}) \in \{0, 1\}$ of $T$ on an input instance $\boldsymbol{x}$ is given by the label of the leaf reached from the root as follows: at each node, go to the left or right child depending on whether the input value of the corresponding variable is $0$ or $1$, respectively. The size of $T$, denoted $|T|$, is given by the number of its nodes.*

The class of decision trees over $X_n$ is denoted $\mathtt{DT}_n$. It is well-known that any decision tree $T \in \mathtt{DT}_n$ can be transformed in linear time into an equivalent disjunction of terms, denoted $\mathtt{DNF}(T)$, where each term corresponds to a path from the root to a leaf labeled with 1. Dually, $T$ can be transformed in linear time into a conjunction of clauses, denoted $\mathtt{CNF}(T)$, where each clause is the negation of the term describing a path from the root to a leaf labeled with 0.

For illustration, the following toy example will be used throughout the paper as a running example:

**Example 1.** *The decision tree in Figure 1 separates Cattleya orchids from other orchids using the following features: $x_1$: "has fragrant flowers", $x_2$: "has one or two leaves", $x_3$: "has large flowers", and $x_4$: "is sympodial".*

As a salient characteristic, decision trees convey a single explicit abductive explanation for classifying any input instance:

**Definition 2** (Direct Reason). *Let $T \in \mathtt{DT}_n$ and $\boldsymbol{x} \in \{0,1\}^n$. The* direct reason *for $\boldsymbol{x}$ given $T$ is the term, denoted $t_{\boldsymbol{x}}^T$, corresponding to the unique root-to-leaf path of $T$ that is compatible with $\boldsymbol{x}$.*

Another important notion of abductive explanations is the following concept of *sufficient reason*[7], that, unlike the notion of direct reason, is not specific to decision trees:

**Definition 3** (Sufficient Reason). *Let $f \in \mathcal{F}_n$ and $\boldsymbol{x} \in \{0,1\}^n$ such that $f(\boldsymbol{x}) = 1$ (resp. $f(\boldsymbol{x}) = 0$). A* sufficient reason *for $\boldsymbol{x}$ given $f$ is a prime implicant $t$ of $f$ (resp. $\neg f$) that covers $\boldsymbol{x}$. $sr(\boldsymbol{x}, f)$ denotes the set of sufficient reasons for $\boldsymbol{x}$ given $f$.*

Thus, a sufficient reason [7] (also known as prime implicant explanation [29]) for an instance $\boldsymbol{x}$ given a class described by a Boolean function $f$ is a subset $t$ of the characteristics of $\boldsymbol{x}$ that is minimal w.r.t. set inclusion such that any instance $\boldsymbol{x}'$ sharing this set $t$ of characteristics is classified by $f$ as $\boldsymbol{x}$ is. Thus, when $f(\boldsymbol{x}) = 1$, $t$ is a sufficient reason for $\boldsymbol{x}$ given $f$ if and only if $t$ is a prime implicant of $f$ such that $\boldsymbol{x}$ implies $t$, and when $f(\boldsymbol{x}) = 0$, $t$ is a sufficient reason for $\boldsymbol{x}$ given $f$ if and only if $t$ is a prime implicant of $\neg f$ such that $t$ covers $\boldsymbol{x}$. Accordingly, sufficient reasons are suited to explain why the instance at hand $\boldsymbol{x}$ has been classified by $f$ as it has been classified. Unlike direct reasons [17], sufficient reasons do not contain any redundant feature.

When considering the sufficient reasons of the input instance, one may be interested in focusing on the shortest ones, alias the minimal sufficient reasons. Those reasons are valuable since conciseness is often a desirable property of explanations (Occam's razor). Formally:

**Definition 4** (Minimal Sufficient Reason). *Let $f \in \mathcal{F}_n$ and $\boldsymbol{x} \in \{0,1\}^n$. A* minimal sufficient reason *for $\boldsymbol{x}$ given $f$ is a sufficient reason for $\boldsymbol{x}$ given $f$ that contains a minimal number of literals.*

Finally, unlike direct and (possibly minimal) sufficient reasons that aim to explain the classification of the instance $\boldsymbol{x}$ under consideration as achieved by the classifier $f$, contrastive explanations are valuable when $\boldsymbol{x}$ has not been classified by $f$ as expected by the explainee. In this case, one looks for minimal subsets of the features that when switched in $\boldsymbol{x}$ are enough to get instances that are classified positively (resp. negatively) by $f$ if $\boldsymbol{x}$ is classified negatively (resp. positively) by $f$. Formally, a *contrastive explanation* for $\boldsymbol{x}$ given $f$ [15] is a subset $t$ of the characteristics of $\boldsymbol{x}$ that is minimal w.r.t. set inclusion among those such that at least one instance $\boldsymbol{x}'$ that coincides with $\boldsymbol{x}$ except on the characteristics from $t$ is not classified by $f$ as $\boldsymbol{x}$ is.

**Definition 5** (Contrastive Explanation). *Let $f \in \mathcal{F}_n$ and $\boldsymbol{x} \in \{0,1\}^n$ such that $f(\boldsymbol{x}) = 1$ (resp. $f(\boldsymbol{x}) = 0$). A* contrastive explanation *for $\boldsymbol{x}$ given $f$ is a term $t$ over $X_n$ such that $t \subseteq t_{\boldsymbol{x}}$, $t_{\boldsymbol{x}} \setminus t$ is not an implicant of $f$ (resp. $\neg f$), and for every $\ell \in t$, $t \setminus \{\ell\}$ does not satisfy this last condition.*

**Example 2.** *Based on our running example, we can observe that $T(\boldsymbol{x}) = 1$ for the instance $\boldsymbol{x} = (1,1,1,1)$. The direct reason for $\boldsymbol{x}$ given $T$ is the term $t_{\boldsymbol{x}}^T = x_1 \wedge x_2 \wedge x_3 \wedge x_4$. $x_1 \wedge x_4$ and $x_2 \wedge x_3 \wedge x_4$ are the sufficient reasons for $\boldsymbol{x}$ given $T$. $x_1 \wedge x_4$ is the unique minimal sufficient reason for $\boldsymbol{x}$ given $T$. $x_4$, $x_1 \wedge x_2$, and $x_1 \wedge x_3$ are the contrastive explanations for $\boldsymbol{x}$ given $T$. Thus, the instance $(1,1,1,0)$ that differs with $\boldsymbol{x}$ only on $x_4$ is not classified by $T$ as $\boldsymbol{x}$ is ($(1,1,1,0)$ is classified as a negative instance).*

We mention in passing that when dealing with decision trees $T$, we could have focused only on explanations for the *positive* instances $\boldsymbol{x}$ given $T$. This comes from the fact that $\mathtt{DT}_n$ is closed under negation, in the sense that for any $T \in \mathtt{DT}_n$, $\neg T$ can be obtained by just replacing from $T$ the label of each leaf with its complement. So, for any instance $\boldsymbol{x} \in \{0,1\}^n$, a direct reason (resp. sufficient reason, minimal sufficient reason, contrastive explanation) explaining why $T(\boldsymbol{x}) = 0$ is precisely the same as a direct reason (resp. sufficient reason, minimal sufficient reason, contrastive explanation) explaining why $(\neg T)(\boldsymbol{x}) = 1$. Considering $T$ or its negation $\neg T$ has no computational impact since $\neg T$ can be computed in time linear in the size of $T$.

## 3  Computing All Sufficient Reasons

**Sufficient reasons can be exponentially numerous.**  When switching from the direct reason for an instance (that is unique but not always redundancy-free) to its sufficient reasons, a main obstacle to be dealt with lies in the number of reasons to be considered. Indeed, even for the restricted class

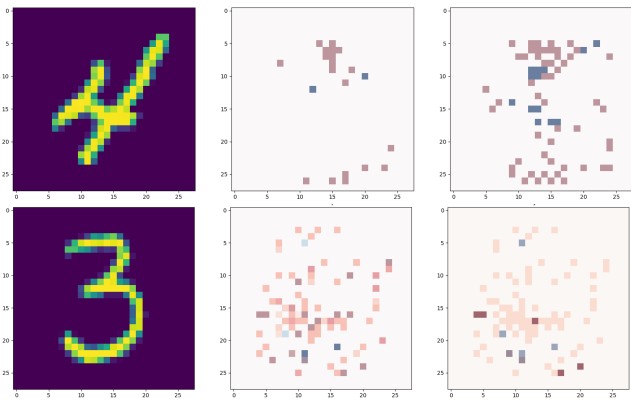

Figure 2: Two sufficient reasons for an `mnist` instance (top), and an explanatory heat map and the explanatory features for an `mnist` instance (bottom).

of decision trees with logarithmic depth, an input instance can have exponentially many sufficient reasons:

**Proposition 1.** *There is a decision tree $T \in \mathtt{DT}_n$ of depth $\log_2(n+1)$ such that for any $\boldsymbol{x} \in \{0,1\}^n$, the number of sufficient reasons for $\boldsymbol{x}$ given $T$ is at least $\lfloor \frac{3}{2}^{\frac{n+1}{2}} \rfloor$.*

By definition, the minimal sufficient reasons for $\boldsymbol{x}$ given $T$ cannot be more numerous than its sufficient reasons. However, focusing on minimal sufficient reasons does not solve the problem since an instance can also have exponentially many minimal sufficient reasons:

**Proposition 2.** *For every $n \in \mathbb{N}$ such that $n$ is odd, there is a decision tree $T \in \mathtt{DT}_n$ of depth $\frac{n+1}{2}$ such that $T$ contains $2n + 1$ nodes and there is an instance $\boldsymbol{x} \in \{0,1\}^n$ such that the number of minimal sufficient reasons for $\boldsymbol{x}$ given $T$ is equal to $2^{\sqrt{n-1}}$.*

In many practical cases, the number of sufficient reasons for an instance given a decision tree can be very large. Figure 2 (top) shows an `mnist` instance (the leftmost subfigure) that has 482 185 073 664 sufficient reasons. Among them there are very dissimilar sufficient reasons. As an illustration, the two rightmost subfigures present two sufficient reasons for this instance, and they differ on many features (blue (resp. red) dots correspond to pixels on (resp. off)).

For such datasets, computing the set of all the sufficient reasons for a given instance is not always feasible. Furthermore, if the computation succeeds but the number of sufficient reasons is huge, their (disjunctively interpreted) set, alias the complete reason for the instance [7], can hardly be considered as intelligible by the explainee. Finally, due to the number of sufficient reasons and their diversity, deriving one of them is not informative enough. Thus, one needs to design approaches to synthesizing their set while avoiding the two pitfalls (the computational one and the informational one).

**Synthesizing the set of sufficient reasons.** In this objective, the following notions of *necessary / (ir)relevant features* appear useful. These notions of necessity and relevance echo the ones that have been considered in [9] for logic-based abduction.

**Definition 6** (Explanatory Features)**.** *Let $f \in \mathcal{F}_n$, and $\boldsymbol{x} \in \{0,1\}^n$ be an instance. Let $e$ be an explanation type.[1]*

- *A literal $\ell$ over $X_n$ is a* necessary feature *for the family $e$ of explanations for $\boldsymbol{x}$ given $f$ if and only if $\ell$ belongs to every explanation $t$ for $\boldsymbol{x}$ given $f$ such that $t$ is of type $e$. $Nec_e(\boldsymbol{x}, f)$ denotes the set of all necessary features for the family $e$ of explanations for $\boldsymbol{x}$ given $f$.*

- *A literal $\ell$ over $X_n$ is a* relevant feature *for the family $e$ of explanations for $\boldsymbol{x}$ given $f$ if and only if $\ell$ belongs to at least one explanation $t$ for $\boldsymbol{x}$ given $f$ such that $t$ is of type $e$. $Rel_e(\boldsymbol{x}, f)$ denotes the set of all relevant features for the family $e$ of explanations for $\boldsymbol{x}$*

---

[1]For instance, $e$ can be $s$ when the sufficient reasons for $\boldsymbol{x}$ given $f$ are targeted or $c$ when the contrastive explanations for $\boldsymbol{x}$ given $f$ are targeted.

*given $f$. $Irr_e(\boldsymbol{x}, f)$, which is the complement of $Rel_e(\boldsymbol{x}, f)$ in the set of all literals over*
*$X_n$, denotes the set of all irrelevant features for the family $e$ of explanations for $\boldsymbol{x}$ given $f$.*

The necessary (resp. irrelevant) features for the family $s$ of sufficient reasons for $\boldsymbol{x}$ given $f$ are the most (resp. less) important features for explaining the classification of $\boldsymbol{x}$ by $f$, since they belong to every (resp. no) sufficient reason for $\boldsymbol{x}$ given $f$.

When a single sufficient reason $t$ for $\boldsymbol{x}$ given $f$ has been computed, the cardinality of $t$ deprived from the features of $Nec_s(\boldsymbol{x}, f)$ is small, and the cardinality of the symmetric difference between $t$ and $Rel_s(\boldsymbol{x}, f)$ is small as well, $t$ can be viewed as a good representative of the complete reason for $\boldsymbol{x}$ given $f$ in the sense that a sufficient reason $t'$ for $\boldsymbol{x}$ given $f$ that differs a lot from $t$ cannot exist.

In the case when $f$ is a decision tree $T$, though the set of all sufficient reasons for $\boldsymbol{x}$ given $T$ cannot be generated when it is too large, $Nec_s(\boldsymbol{x}, f)$, $Rel_s(\boldsymbol{x}, f)$, and $Irr_s(\boldsymbol{x}, f)$ can be derived efficiently:

**Proposition 3.** *Let $T \in \mathtt{DT}_n$, and $\boldsymbol{x} \in \{0, 1\}^n$. Computing $Nec_s(\boldsymbol{x}, T)$, $Rel_s(\boldsymbol{x}, f)$, and $Irr_s(\boldsymbol{x}, T)$ can be done in $\mathcal{O}((n + |T|) \times |T|)$ time.*

Going a step further consists in evaluating the explanatory importance of every (positive or negative) feature:

**Definition 7** (Explanatory Importance)**.** *Let $f \in \mathcal{F}_n$, and $\boldsymbol{x} \in \{0, 1\}^n$ be an instance. Let $e$ be an explanation type, and $E_e(\boldsymbol{x}, f)$ the set of all explanations for $\boldsymbol{x}$ given $f$ that are of type $e$. The* explanatory importance *of a literal $\ell$ over $X_n$ for $\boldsymbol{x}$ given $f$ w.r.t. $e$ is given by*

$$Imp_e(\ell, \boldsymbol{x}, f) = \frac{\#(\{t \in E_e(\boldsymbol{x}, f) : \ell \in t\})}{\#(E_e(\boldsymbol{x}, f))}.$$

**Example 3.** *On the running example, we have $Nec_s(\boldsymbol{x}, T) = \{x_4\}$, and $Rel_s(\boldsymbol{x}, T) = \{x_1, x_2, x_3, x_4\}$. We also have $Imp_s(x_4, \boldsymbol{x}, T) = 1$, $Imp_s(x_1, \boldsymbol{x}, T) = Imp_s(x_2, \boldsymbol{x}, T) = Imp_s(x_3, \boldsymbol{x}, T) = \frac{1}{2}$, and $Imp_s(\ell, \boldsymbol{x}, T) = 0$ for every other literal $\ell$ (the negative ones over $\{x_1, x_2, x_3, x_4\}$).*

The notion of explanatory importance must not be confused with the notions of feature importance (which can be defined and assessed in many different ways): the former is local (i.e., relative to an instance) and not global, it concerns literals and not variables (polarity matters), and it is about the explanation task, not the prediction one.

In order to compute the explanatory importance of a literal, a straightforward approach consists in enumerating the explanations of $E_e(\boldsymbol{x}, f)$. This is feasible when this set is not too large, which is not always the case for sufficient reasons even when $f$ is a decision tree $T$. Thus, for dealing with the remaining case, an alternative approach must be looked for.

We designed such an approach for computing $Imp_s(\ell, \boldsymbol{x}, T)$. We know that $sr(\boldsymbol{x}, T)$ is by construction the set of prime implicants of $g = \{c \cap t_{\boldsymbol{x}} : c \in \mathtt{CNF}(T)\}$. Thus, we exploited the translation presented in [18] showing how to associate in polynomial time with a given $\mathtt{CNF}$ formula (here, $g$) another formula (over a distinct set of variables), let us say $h$, such that the models of $h$ are in one-to-one correspondence with the prime implicants of $g$. In our case, the translation can be simplified because $g$ is a monotone $\mathtt{CNF}$ formula. Since $h$ is not primarily a $\mathtt{CNF}$ formula, leveraging Tseitin transformation [31], we turned $h$ in linear time into a query-equivalent $\mathtt{CNF}$ formula $i$. Note that every auxiliary variable that is introduced in $i$ is defined from the other variables (those occurring in $h$), so that the number of models of $i$ is the same as the number of models of $h$. Finally, we took advantage of the compilation-based model counter $\mathtt{D4}$ [20] to compile $i$ into a $\mathtt{d\text{-}DNNF}$ circuit [6], and this enabled us to compute in time polynomial in the size of $i$ both the number of sufficient reasons and the explanatory importance of every literal (indeed, the $\mathtt{d\text{-}DNNF}$ language supports in polytime the model counting query and the conditioning transformation [8]). We show in Section 5 that, despite a high complexity in the worst case (the size of $i$ can be exponential in $|T|$), this approach based on knowledge compilation proves quite efficient in practice.

Clearly enough, when $Imp_e(\ell, \boldsymbol{x}, T)$ has been computed for every $\ell$, one can easily generate explanatory heat maps. Figure 2 (bottom) shows an $\mathtt{mnist}$ instance (the leftmost subfigure) that has 19 115 685 sufficient reasons, 6 necessary literals, and 94 relevant literals. The central subfigure is the corresponding heat map. Blue (resp. red) pixels correspond to positive (resp. negative) literals in the instance, and the intensity of the color aims to reflect the explanatory importance of the corresponding literal. The rightmost subfigure gives the explanatory features (dark pixels are associated with necessary literals, and light pixels to relevant literals).

**Enumerating the minimal sufficient reasons.** An approach to synthesizing the set of sufficient reasons consists in focusing on the minimal ones. Indeed, though the set of minimal sufficient reasons for an instance given a decision tree can be exponentially large, the number of minimal sufficient reasons cannot exceed the number of sufficient reasons, and it can be significantly lower in practice.

However, unlike sufficient reasons that can be generated in polynomial time using a greedy algorithm (see e.g., [17]), computing minimal reasons is not an easy task:

**Proposition 4.** *Let $T \in \mathtt{DT}_n$ and $\boldsymbol{x} \in \{0,1\}^n$. Computing a minimal sufficient reason for $\boldsymbol{x}$ given $T$ is NP-hard.*

Despite this intractability result, minimal sufficient reasons can be generated in many practical cases. A common approach for handling NP-optimization problems is to rely on modern constraint solvers. One follows this direction here and casts the task of finding minimal sufficient reasons as a Boolean constraint optimization problem. We first need to recall that a PARTIAL MAXSAT problem consists of a pair $(C_{\mathrm{soft}}, C_{\mathrm{hard}})$ where $C_{\mathrm{soft}}$ and $C_{\mathrm{hard}}$ are (finite) set of clauses. The goal is to find a Boolean assignment that maximizes the number of clauses $c$ in $C_{\mathrm{soft}}$ that are satisfied, while satisfying all clauses in $C_{\mathrm{hard}}$.

**Proposition 5.** *Let $T$ be a decision tree in $\mathtt{DT}_n$ and $\boldsymbol{x} \in \{0,1\}^n$ be an instance such that $T(\boldsymbol{x}) = 1$. Let $(C_{\mathrm{soft}}, C_{\mathrm{hard}})$ be an instance of the PARTIAL MAXSAT problem such that:*

$$C_{\mathrm{soft}} = \{\overline{x_i} : x_i \in t_{\boldsymbol{x}}\} \cup \{x_i : \overline{x_i} \in t_{\boldsymbol{x}}\} \text{ and } C_{\mathrm{hard}} = \{c \cap t_{\boldsymbol{x}} : c \in \mathtt{CNF}(T)\}.$$

*The intersection of $t_{\boldsymbol{x}}$ with $t_{\boldsymbol{x}^*}$ where $\boldsymbol{x}^*$ is an optimal solution of $(C_{\mathrm{hard}}, C_{\mathrm{soft}})$, is a minimal sufficient reason for $\boldsymbol{x}$ given $T$.*

Clearly enough, if $\boldsymbol{x}$ is such that $T(\boldsymbol{x}) = 0$, then it is enough to consider the same instance of PARTIAL MAXSAT as above, except that $C_{\mathrm{hard}} = \{c \cap t_{\boldsymbol{x}} : c \in \mathtt{CNF}(\neg T)\}$.

Finally, one can take advantage of this PARTIAL MAXSAT characterization for generating a preset number of minimal sufficient reasons (basically, one generates a first reason $t$, then one adds to $C_{\mathrm{hard}}$ the negation of $t$ as a clause as well as a CNF encoding of a cardinality constraint for ensuring that the next reasons to be generated have the same size as the one of $t$, and we resume until the bound is reached or no solution exists).

# 4 Computing All Contrastive Explanations

Interestingly, it has been shown that sufficient reasons and contrastive explanations are connected by a minimal hitting set duality [15]. This duality can be leveraged to derive one of the two sets of explanations from the other one using algorithms for computing minimal hitting sets [27, 32].

However, in the case of decision trees, a more direct and much more efficient approach to derive all the contrastive explanations for $\boldsymbol{x} \in \{0,1\}^n$ given $T \in \mathtt{DT}_n$ can be designed. Indeed, unlike what happens for sufficient reasons (see Section 3), the set of *all* contrastive explanations for $\boldsymbol{x} \in \{0,1\}^n$ given a decision tree $T \in \mathtt{DT}_n$ can be computed in polynomial time from $\boldsymbol{x}$ and $T$:

**Proposition 6.** *The set of all contrastive explanations for $\boldsymbol{x} \in \{0,1\}^n$ given a decision tree $T \in \mathtt{DT}_n$ can be computed in time polynomial in $n + |T|$ as $min(\{c \cap t_{\boldsymbol{x}} : c \in \mathtt{CNF}(f)\}, \subseteq)$.*

**Example 4.** *On the running example, we have $\mathtt{CNF}(T) = \{x_1 \vee x_2, x_1 \vee \overline{x_2} \vee x_3, x_1 \vee \overline{x_2} \vee \overline{x_3} \vee x_4, \overline{x_1} \vee x_2 \vee x_3 \vee x_4, \overline{x_1} \vee x_2 \vee \overline{x_3} \vee x_4, \overline{x_1} \vee \overline{x_2} \vee x_3 \vee x_4, \overline{x_1} \vee \overline{x_2} \vee \overline{x_3} \vee x_4\}$. Thus, with $\boldsymbol{x} = (1,1,1,1)$, we have $min(\{c \cap t_{\boldsymbol{x}} : c \in \mathtt{CNF}(f)\}, \subseteq) = \{x_1 \vee x_2, x_1 \vee x_3, x_4\}$, which corresponds to the contrastive explanations $x_1 \wedge x_2$, $x_1 \wedge x_3$, $x_4$ for $\boldsymbol{x}$ given $T$ (viewing clauses and terms as sets of literals).*

As straightforward consequences of Proposition 6, computing necessary / relevant features and computing the explanatory importance of features w.r.t. contrastive explanations can be achieved in time polynomial in $n + |T|$. Similarly, statistics about the size of contrastive explanations can be easily established, and contrastive explanations can be easily minimized and counted.

# 5 Experiments

**Empirical setting.** We have considered 90 datasets, which are standard benchmarks from the well-known repositories Kaggle (`www.kaggle.com`), OpenML (`www.openml.org`), and UCI (`archive.`

Table 1: Empirical results based on 12 datasets.

| Dataset | Decision Tree | | | |Sufficient| | | |Minimal| | | #Nec. Features | | #Rel. Features | |
|---|---|---|---|---|---|---|---|---|---|---|---|
| | %A | #N | #B | med | max | med | max | med | max | med | max |
| recidivism | 63.41 | 13828.80 | 147.60 | 14 | 22 | 13 | 22 | 6 | 19 | 60 | 98 |
| adult | 81.36 | 12934.00 | 2974.80 | 16 | 36 | 16 | 36 | 7 | 22 | 263 | 543 |
| bank marketing | 87.40 | 6656.40 | 1432.60 | 14 | 21 | 14 | 21 | 3 | 16 | 247 | 398 |
| bank | 88.99 | 5523.60 | 977.80 | 13 | 24 | 13 | 24 | 4 | 15 | 200 | 330 |
| lending loan | 73.49 | 2610.40 | 1131.40 | 16 | 31 | 16 | 31 | 8 | 25 | 226 | 442 |
| contraceptive | 50.44 | 1252.20 | 88.60 | 11 | 20 | 11 | 20 | 8 | 17 | 25 | 47 |
| compas | 65.98 | 1230.00 | 46.20 | 6 | 14 | 6 | 14 | 3 | 12 | 16 | 33 |
| christine | 63.36 | 853.20 | 426 | 12 | 47 | 12 | 47 | 8 | 41 | 92 | 202 |
| farm-ads | 86.75 | 544.80 | 264.60 | 20 | 99 | 20 | 99 | 16 | 92 | 73 | 192 |
| mnist49 | 95.47 | 539.60 | 267.90 | 22 | 30 | 22 | 30 | 9 | 19 | 91 | 166 |
| spambase | 91.94 | 536.40 | 264.80 | 15 | 29 | 15 | 29 | 9 | 24 | 68 | 146 |
| mnist38 | 96.07 | 506.60 | 251.40 | 19 | 28 | 19 | 28 | 8 | 20 | 93.50 | 157 |

| Dataset | #Sufficient | | #Contrastive | | |Contrastive| | | #Minimal | |
|---|---|---|---|---|---|---|---|---|
| | med | max | med | max | med | max | med | max |
| recidivism | 10387 | 9734080 | 54 | 145 | 3 | 16 | 2 | 144 |
| adult | - | $\geq$ 15738357226073000000000 | 201 | 470 | 4 | 16 | 3 | 256 |
| bank marketing | - | $\geq$ 7460375213484350000000 | 189 | 337 | 4 | 13 | 8 | 432 |
| bank | - | $\geq$ 74339519790185000000 | 150 | 277 | 4 | 13 | 4 | 168 |
| lending loan | 459258918095775 | 943243242816203000000000000000 | 157 | 311 | 3 | 12 | 3 | 192 |
| contraceptive | 20,50 | 4272 | 21 | 52 | 2 | 11 | 2 | 48 |
| compas | 16 | 444 | 13 | 33 | 2 | 11 | 2 | 21 |
| christine | 63108 | 2167735434744 | 71 | 151 | 3 | 8 | 2 | 4096 |
| farm-ads | 1177,50 | 921895392 | 59 | 166 | 2 | 10 | - | $\geq$ 10000 |
| mnist49 | 7392384 | 715892613696000 | 61 | 106 | 2 | 12 | - | $\geq$ 10000 |
| spambase | 15712 | 2535069312 | 50 | 107 | 2 | 11 | 4 | 384 |
| mnist38 | 14849376 | 16922386736640 | 62 | 107 | 3 | 11 | 32 | 3072 |

ics.uci.edu/ml/). `mnist38` and `mnist49` are subsets of the `mnist` dataset, restricted to the instances of 3 and 8 (resp. 4 and 9) digits. Because some datasets are suited to the multi-label classification task, we used the standard "one versus all" policy to deal with them: all the classes but the target one are considered as the complementary class of the target. Categorical features have been treated as arbitrary numbers (the scale is nominal). As to numeric features, no data preprocessing has taken place: these features have been binarized on-the-fly by the decision tree learning algorithm that has been used.

For every benchmark $b$, a 10-fold cross validation process has been achieved. Namely, a set of 10 decision trees $T_b$ have been computed and evaluated from the labelled instances of $b$, partitioned into 10 parts. One part was used as the test set and the remaining 9 parts as the training set for generating a decision tree. This tree is thus in 1-to-1 correspondence with the test set chosen within the whole dataset $b$. The classification performance for $b$ was measured as the mean accuracy obtained over the 10 decision trees generated from $b$. The CART algorithm, and more specifically its implementation provided by the Scikit-Learn library [25] has been used to learn decision trees. All hyper-parameters of the learning algorithm have been set to their default value. Notably, decision trees have been learned using the Gini criterion, and without any maximal depth or any other manual limitation.

For each benchmark $b$, each decision tree $T_b$, and a subset of at most 100 instances $x$ picked up at random in the test set following a uniform distribution, we computed a sufficient reason for $x$ given $T_b$ (using the standard greedy algorithm run on the direct reason $t_x^{T_b}$), and a minimal sufficient reason for $x$ given $T_b$ using the PARTIAL MAXSAT encoding presented in Proposition 5. This enabled us to draw some statistics (median, maximum) about the sizes of the reasons that have been generated. Using the algorithm presented in the proof of Proposition 3, we also derived the necessary and relevant explanatory features for each $x$, and again drew some statistics about them. Exploiting the model counter D4, we computed the number of sufficient reasons for $x$ given $T_b$, as well as the explanatory importance of every feature. Taking advantage of the algorithm given in Proposition 4, we computed the number of contrastive explanations for $x$ given $T_b$, and drew some statistics about those numbers and about the sizes of the contrastive explanations. Finally, using the approach described in Section 3, we enumerated all the minimal sufficient reasons for $x$ given $T_b$ up to a limit of 10 000, and again drew some statistics about the numbers of minimal sufficient reasons. Of course, for each computation, we measured the corresponding runtimes since this is fundamental to determine the extent to which the algorithms are practical (details are provided as a supplementary material).

All the experiments have been conducted on a computer equipped with Intel(R) XEON E5-2637 CPU @ 3.5 GHz and 128 GiB of memory. D4 [20] was run with its default parameters. For computing

minimal reasons, we used the Pysat library [14], which provides the implementation of the RC2 PARTIAL MAXSAT solver. This solver was run using the parameters corresponding to the "Glucose" setting. A time-out of 100s per instance was set for D4.

**Results.** Table 1 (top and bottom) reports an excerpt of our results, focusing on 12 benchmarks out of 90 (the selected datasets are among those containing many instances and/or many features). The leftmost column gives the name of the dataset $b$. Columns $\%A$, $\%N$, and $\#B$ give, respectively, the mean accuracy over the 10 decision trees, the average number of nodes in those trees, and the average number of binary features they are based on. The next columns give statistics (median, maximum) about, respectively, the size of the sufficient reasons (|Sufficient|) and of the minimal sufficient reasons (|Minimal|) that have been computed, as well as about the number of necessary (#Nec. Features) and relevant (#Rel. Features) features that appear in the full set of sufficient reasons for the instance. Table 1 (bottom) give statistics (median, maximum) about, respectively, the number of sufficient reasons (#Sufficient), the number of contrastive explanations (#Contrastive) and their sizes (|Contrastive|), and finally the number of minimal sufficient reasons (#Minimal).

As to the computation times, it turns out that all the algorithms described in the previous sections proved as efficient in practice. This is not surprising for those algorithms having a polytime worst-case complexity (the greedy algorithm for computing a sufficient reason, the one for deriving explanatory features, and the one for computing all the contrastive explanations). It was less obvious at first sight for the algorithms used for counting the number of sufficient reasons and for computing the explanatory importance of features. However, all the computations that have been run have terminated in due time, except for 3 datasets out of 90, namely `adult`, `bank_marketing`, and `bank`. For these datasets, the time limit of 100s has been reached for, respectively, 203, 150, and 336 instances out of 1000 (in this case, the median number of sufficient reasons has not been reported). Notably, for all the 90 datasets but those 3, the median time required for counting the number of sufficient reasons and computing the explanatory importance of features never exceeded 1s. Computing a minimal sufficient reason, and more generally all such reasons looked challenging as well, due to both the intrinsic complexity of computing a minimal sufficient reason and to their number. Nevertheless, our enumeration algorithm succeeded in deriving *all the minimal sufficient reasons* for every dataset except 3 out of 90, namely `farm-ads`, `mnist49`, and `gisette`. For these datasets, the limit of 10 000 reasons has been reached for, respectively, 5, 16, and 3 instances out of 1000. Interestingly, the median time needed to derive all the minimal sufficient reasons for the instances for which the computation has been successful exceeded 1s only for 2 datasets (`adult` and `bank_marketing`).

Beyond providing evidence that the number of reasons can be huge, our experiments have highlighted that the greedy algorithm for deriving a sufficient reason computes in practice a minimal sufficient reason in many cases. They have also shown that the number of explanatory relevant features for an instance is typically much lower than the number of binary features used to describe it, and that the number of explanatory necessary features is also significantly lower than the number of explanatory relevant features. The gap between the two explains the possibly enormous number of sufficient reasons. When considering the full set of reasons, a considerable difference between the number of sufficient reasons and the number of minimal sufficient reasons can also be observed. Finally, like minimal sufficient reasons, the number of contrastive explanations appears in many cases not very large, which is a good point from an intelligibility perspective.

## 6   Conclusion

In light of our results, it turns out that the explanatory power of decision trees goes far beyond its ability to generate direct reasons. From a decision tree, the explanatory importance of features and the minimal sufficient reasons for an instance can be computed efficiently most of the time. For decision trees, fully addressing the "Why not?" question also appears as easier than fully addressing the "Why?" question: computing the full set of sufficient reasons for the instance at hand is typically out of reach, while computing its full set of contrastive explanations is tractable.

Accordingly, the language of decision trees appears not only as appealing for the learning purpose, but also as a good target when one needs to reason on the various forms of explanations (abductive and contrastive ones) associated with the predictions made. This coheres with (and completes) the results reported in [1], showing that many other explanation and verification tasks are tractable for decision tree classifiers.

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
