# OpenReview forum: "On the Explanatory Power of Decision Trees"
_NeurIPS.cc/2021/Conference — NeurIPS 2021 Submitted_

### Official Review · Reviewer_7zMv · 2021-07-13

**Rating:** 3
**Confidence:** 3

**Summary:**

The paper argues that information other than what is presented explicitly in a decision tree can be extracted, in some cases efficiently, that is better at explaining the decisions taken according to the decision tree. The paper proceeds to show some statistics about the type of information that is argued for, when considering various datasets.


**Ethical Concerns:**

Not something obvious.


**Limitations And Societal Impact:**

Seems fine.


**Main Review:**

Ultimately, this is a paper about properties of classical logic and prime implicants. It has nothing to do with a conference on machine learning. Also, just by reading the introduction one can see that the claims made are not surprising, follow from very basic proof techniques, and are probably established in some form or another in any textbook on classical logic for undergraduate students.

To complicate matters further, the terminology used in the paper is non-standard. E.g., using "sufficient reason" for a prime implicant, and then using "minimal sufficient reason" to talk about the size of the implicant (not its set-theoretic minimality) is against the intuitive meaning of the concepts "sufficiency" and "minimal sufficiency", which naturally correspond to implicants and prime implicants.

The motivation of the paper is also awkward. It first discusses explainability and decision trees to motivate the use of prime implicants, but then shows a mnist dataset and talks about explanations in terms of pixels. Do the authors suggest that one would learn a decision tree with the the pixels being the features? This confounding of decision trees and pixel-based examples seems to run throughout the paper. Perhaps the discussion of decision trees should be dropped altogether (and also the results about computing prime implicants, which is not a novel contribution), and the paper should focus on the introduction of the metric on explanatory power and its empirical evaluation?

In line with the above, I was expecting the empirical study to give some evidence in support of an otherwise ad hoc definition of explanatory power, but unfortunately the empirical study was an (unneeded) confirmation that formulas can have numerous prime implicants, but few size-minimal prime implicants.

I also found that the paper might be presenting false dilemmas in attempting to motivate things; e.g., the intractability of computing all sufficient reasons, or the intractability of computing even a single minimal sufficient reason. If one wishes to compute all the sufficient reasons, why not simply make do with the path of the decision tree? Is there any evidence that humans find the former more appropriate than the latter as a reasonable explanation? As for the minimal sufficient reason, what evidence is there that minimality (which is in the size of the reason, not in the set theoretic sense) is warranted? I was under the impression, but I could be wrong, that from the early days of AI people realized that size-minimality is a bad compass when judging abductive proofs.

Even if there is something interesting from page 6 onwards, the first half of the paper is sufficiently trivial to diminish the value of the paper.

--- after reading the author response ---

I would like to thank the authors for their response. I am still not clear on certain aspects of the paper.

Clearly, using logic for explainability of ML is relevant to the conference. I didn't mean to suggest otherwise. But this paper is about properties of classical logic. The connection to explainability is not that evident. In particular, regarding the point on whether minimality of prime implicants is a reasonable requirement to impose on the explanations, I wonder: if A by itself explains the label of a data point, and B&C together explain the label of that same data point, does minimality suggest that the first explanation is better than the second?

What if the label to predict is whether someone can repay their loan, and feature A is "they live in the suburbs", whereas features B&C are "they have low income & they have other loans". Is it still the case that A is a better explanation than B&C (assuming that both are indeed prime implicants)? As I said in my review, size-minimality is not necessarily what one might be looking for. Maybe it makes sense for the particular application domain (with features being pixels), but the theory developed in this paper is not specific to such features (on the contrary, given the use of DTs I would say it is more geared towards domains amenable to traditional learning and feature-engineering).

I quickly scanned the two papers cited by the authors in their response. Indeed, at least one of the papers discusses minimum-cardinality as a criterion, but does so as part of a comprehensive look at various minimality criteria that have been considered in the literature. That paper mentions conditions under which every minimality criterion is applicable, and their goal is not to take sides. In the submitted paper, on the other hand, the authors do take a side, and they need to argue why minimal-cardinality is the right approach. Just claiming that past works did not disqualify size-minimal abductive proofs is quite different from disqualifying all non-size-minimal proofs.

Just because the use of the results in this paper are for the purposes of XAI, this does not make them novel. Establishing bounds on the number of prime implicants, the complexity of computing them, etc., are questions in logic, and even if classical logic textbooks do not talk about XAI, they do talk about bounds on prime implicants and their complexity (including in the papers cited by the authors).

The response of the authors that minimality is good for explainability because "a sufficient reason that includes hundreds of features can hardly be considered as intelligible" seems to present yet another false dilemma: is minimality the opposite of hundreds of features? What if a minimal sufficient reason did indeed have hundred of features? Would the minimality still make it okay? Or what if a non-minimal sufficient reason had just 3 features? Would its non-minimality make it a bad choice (e.g., the B&C example above)?

I feel that all the above need to be clarified, perhaps by rethinking how to position this work: Does this work really have to do with decision-trees and their explanatory power, or with computing minimal sufficient reasons irrespective of what the original model is? What are the implied assumptions on the use of the theory? Are you assuming traditional engineered features for which DTs have been used, or unstructured data and images? Perhaps narrowing down a bit the scope of this work would give it more traction.


**Time Spent Reviewing:**

3-4

---

> ### Author Response · Authors · 2021-08-10
> **Response to the review by Reviewer 7zMv**
>
> Thanks for your comments.
>
> _''Ultimately, this is a paper about properties of classical logic and prime implicants. It has nothing to do with a conference on machine learning.’’_
>
> For sure, the paper is not a pure ML paper. However, the CFP of NeurIPS’21 explicitly states that papers about ''Social Aspects of Machine Learning (e.g., AI  safety, fairness, privacy, interpretability)’’ are welcome. Furthermore, decision trees  are a well-known ML model for decades. Finally, please note that many papers about  model-accurate, formal explanations have been published in ML conferences (including ICML and NeurIPS) for the past few years.
>
> _''the claims made are not surprising … are probably established in some form or  another in any textbook on classical logic for undergraduate students.’’_
>
> We respectfully disagree with this comment; as far as we know, no textbook on classical logic is concerned with the XAI issues examined in the present paper. Furthermore, in our opinion, most of the theoretical results reported in the paper are not straightforward.
>
> _''the terminology used in the paper is non-standard. E.g., using "sufficient reason" for a prime implicant’’._
>
> Please note that a sufficient reason is not any prime implicant of the function at hand (it  depends on the instance to be explained since it must cover it). Furthermore, the terminology we used is the standard one (the definition of the notion of sufficient reason we considered is precisely the one introduced in [8]).
>
> _''It first discusses explainability and decision trees to motivate the use of prime  implicants, but then shows a mnist dataset and talks about explanations in terms of  pixels.’’_
>
> Indeed, for such pictures, features correspond to pixels. We made such a choice of a dataset because it allows for getting visual representations of explanations and of  explanatory features. They are simple and can be grasped easily as a whole by the reader. We are afraid that we do not understand why you found this confounding.  Please note that previous papers following a similar research line as the one used in  this work (i.e., modeling ML models as functionally equivalent circuits in or to achieve  explanation/verification tasks on such circuits) also used digits datasets for the  matter of illustration or experimentation (see [17,34]).
>
> _''the empirical study was an (unneeded) confirmation that formulas can have  numerous prime implicants, but few size-minimal prime implicants.’’_
>
> The conclusion of the empirical study does not reduce to that (among others, results  about numbers of explanatory features and about computation times for deriving  explanations and related notions are provided, please see the last paragraph of Section 5).
>
> _''If one wishes to compute all the sufficient reasons, why not simply make do with the path of the decision tree?’’_
>
> The direct reason of an instance is in general not a sufficient one (it may contain redundant features), and many sufficient reasons that differ significantly from the  direct reason may exist. Focusing on the direct reason is thus not guaranteed to be  the best option.
>
> _''As for the minimal sufficient reason, what evidence is there that minimality (which is in the size of the reason, not in the set theoretic sense) is warranted?’’_
>
> It is warranted by definition. Do you mean ``is expected’’? If so, minimality in terms of  size is good for the sake of intelligibility (a sufficient reason that includes hundreds of  features can hardly be considered as intelligible).
>
> _''I was under the impression, but I could be wrong, that from the early days of AI  people realized that size-minimality is a bad compass when judging abductive  proofs.’’_
>
> We respectfully disagree with this. For instance, in the seminar papers of Bylander et al (AIJ,  1991) and of Eiter & Gottlob (JACM, 1995) about the complexity of abduction,  minimal-sized abductive explanations are not disqualified.

---

> ### Author Response · Authors · 2021-09-11
> **Response to additional remarks**
>
> Thanks a lot for your additional remarks after our first response. We share many of your points, and would like to take advantage of an additional response to clarify our point of view about minimal sufficient reasons. Our claim is not that the minimal sufficient reasons of an instance are intrinsically better explanations than the sufficient reasons of the instance that are not minimal. The example you point out illustrates clearly that this is not the case, and we agree with that. Please note that if this had been our claim, the definitions and results reported in Section 3, and the experiments performed as presented in Section 4 would have been different. Especially, when computing the explanatory importance of features, we would have focused on minimal sufficient reasons. This is not the case (we considered all the sufficient reasons). In our opinion, focusing on the minimal sufficient reasons can be useful for two purposes: on the one hand, their set can be much smaller than the whole set of sufficient reasons, so that in some cases, they can be enumerated and reported to the explainee, while it would not make sense to do it with all sufficient reasons because of their number; on the other hand, minimal sufficient reasons are (by definition) the shortest sufficient reasons, and conciseness is good for intelligibility. Whatever the case, the goal is not guaranteed to be reached: on the one hand, an instance may have exponentially many minimal sufficient reasons (this is precisely stated by our Proposition 2); on the other hand, minimality does not imply intelligibility (minimal sufficient reasons can be very large anyway as you point out). This explains why the stress has not been put on minimal sufficient reasons in the paper but instead, on the concepts of relevant features, necessary features, and explanatory importance of features.
>
> Finally, about the positioning of the results w.r.t. logic and XAI: clearly, the notion of sufficient reason comes from previous work on XAI, and is not specifically relevant to Boolean logic. Of course, sufficient reasons are prime implicants, a well-known concept in logic for decades, but as we mentioned in our previous response, sufficient reasons are prime implicants of a special kind. And it would be wrong to believe that every result about prime implicants could be straightforwardly lifted to sufficient reasons. For instance, using the inductive characterization of prime implicants recalled in the proof of Proposition 2, it is not hard to show that the prime implicants of a decision tree (and more generally of a branching program) can be enumerated in output polynomial time. However, the argument used to show it does not extend to the case of sufficient reasons.

---

### Official Review · Reviewer_DBdD · 2021-07-17

**Rating:** 4
**Confidence:** 5

**Summary:**

The paper states a number of results about computing abductive and
contrastive explanations for decision trees. The paper includes some
experiments on computing explanations about decision trees.


**Ethical Concerns:**

None.

**Limitations And Societal Impact:**

None.

**Main Review:**

The paper states a number of results about abductive explanations and
contrastive explanations of decision tres.

Main claims:

Proposition 1 states a lower bound on the worst-case number of
sufficient reasons for a prediction.

Proposition 2 states a similar results but for the number of minimal
sufficient reasons.

Proposition 3 states that some specific sets of features can be
computed in polynomial time.

Proposition 4 states the NP-hardness of computing a minimal sufficient
reason.

Proposition 5 states the correctness of a partial MaxSAT encoding for
computing a minimal sufficient reason.

Proposition 6 states that contrastive explanations can be computed in
polynomial time.

Additional Comments:

Starting on page 2, the paper delves into 'abductive explanations'.
However, 'abductive explanations' were introduced in [16], but the
paper never relates with this earlier work. My understanding is that
the 'abductive explanations' proposed in [16] are related with
'sufficient reasons' proposed in later work [7] and with
'PI-explanations' proposed in earlier work [29]. The main difference
is that 'PI-explanations' and 'sufficient reasons' are studied in the
context of boolean classifiers and this is not the case with
'abductive explanations'. The paper refers to 'abductive explanations'
as something different, without relating with [16], and does not state
the relationship. This should be corrected.

A similar comment applies to the relationship between abductive and
contrastive explanations and "Why?" and "Why Not?" explanations. This
is discussed in detail in [15]. Granted that the paper cites [15], but
not when introducing contrastive explanations.

The definitions of sufficient reason and contrastive explanations
are much more complicated that other treatments, e.g. [15], or even
more recent papers on the same subject.

It would be important to relate Definition 6 with recent work on
explainability, concretely reference [1].

The paper discusses decision trees with boolean features, but this is
too restrictive. For example, there are situations where features are
real-valued. Also, the encodings of decision trees to DNF and CNF
assume one want to analyze a prediction of 1. This should be
clarified, as this impacts the later results.

The decision not to include the proofs in the paper means that
important information is being left out of the paper. The proofs or at
least the most relevant ones should be included in the paper.

Regarding the results stated in the paper, a few observations can be
made.

Proposition 4 has been proved in earlier work:

[R1] P. Barcelo, M. Monet, J. Perez, B. Subercaseaux: Model
Interpretability through the lens of Computational Complexity. NeurIPS
2020

which the paper does not reference. Concretely, Proposition 4 is one
of the results included in the proof Proposition 6 of the paper
above.

The issue of the encoding described above becomes problematic when
discussing Propositions 5 and 6. It seems to me the paper is focusing
on a prediction of 1. Is that the case? If so, why? It seems to me
there is no need for that, as long as one does not reason in terms of
the encoding.

Regarding Proposition 6 in the paper, the proof exploits the CNF
encoding proposed earlier in the paper. This is unnecessary and makes
the result depend on the value of the prediction. A simpler proof of
the same result is available in the following work:

[R2] X. Huang, Y. Izza, A. Ignatiev, J. Marques-Silva: On Efficiently
Explaining Graph-Based Classifiers. CoRR abs/2106.01350 (2021)

With respect to the experiments, the decision to binarize features
should be explained in detail. Binarization requires the explanation
algorithms to be aware that binarization took place. This is discussed
for example in [R2] above.

Also, although the paper makes a good point that the number of minimal
sufficient explanations is in general much smaller than the number of
sufficient explanations, it offers no evidence that the size of
minimal sufficient explanations is significantly smaller than the size
of sufficient explanations.

Overall, I believe some of the stated results provide interesting
insights about explanations in decision trees. However, the writing is
unnecessarily complicated. Also, the should do a much better job of
relating with earlier work. In addition, some of the results stated in
the paper are already known.

**Time Spent Reviewing:**

3h

---

> ### Author Response · Authors · 2021-08-10
> **Response to the review by Reviewer DBdD**
>
> Thanks for your comments.
>
> _``Starting on page 2, the paper delves into 'abductive explanations'. However,  'abductive explanations' were introduced in [16], but the paper never relates with  this earlier work. … The paper refers to 'abductive explanations' as something  different, without relating with [16], and does not state the relationship. This should  be corrected.’’_
>
> The phrasing was on purpose. Our goal was to keep the meaning of 'abductive  explanations' informal in the introduction (idem for ‘contrastive explanations’).  Indeed, Definition 3 in [16] requires subset minimality, which was considered as a preference criterion in early work about abduction in AI, and not as a condition to be fulfilled by any abductive explanation. We do not want to be that specific here, in order to consider the direct reason of an instance as an abductive explanation, since it is about answering the `Why?’ question. We can easily clarify this, and cite [15,16] as expected.
>
> _``It would be important to relate Definition 6 with recent work on explainability,  concretely reference [1].’’_
>
> The notion of (ir)relevance of a feature considered in [1] concerns classes, not explanations (which correspond to specific instances). This is a significant difference, and we can easily elaborate on it.
>
> _``The paper discusses decision trees with boolean features, but this is too  restrictive. For example, there are situations where features are real-valued.’’_
>
> Indeed, we considered data sets with numeric features in our experiments.
>
> _``Also, the encodings of decision trees to DNF and CNF assume one want to  analyze a prediction of 1. This should be clarified, as this impacts the later results.’’_
>
> Yes, this can be assumed without loss of generality since we can easily negate a  decision tree (please see the last paragraph before Section 3).
>
> _``The decision not to include the proofs in the paper means that important  information is being left out of the paper. The proofs or at least the most relevant  ones should be included in the paper.’’_
>
> Proofs were provided as a supplementary material.
>
> _``Proposition 4 has been proved in earlier work’’._
>
> We missed this paper. Thanks for pointing out the reference.
>
> _``A simpler proof of the same result is available in the following work’’._
>
> Thanks again for pointing out this reference. However, please note that the NeurIPS’21 FAQ indicates that ``authors are not expected to compare to work that appeared only a month or two before the deadline’’ and that reference (CoRR abs/2106.01350) has been uploaded on arXiv by June 2nd, 2021 … thus **after** the submission deadline for NeurIPS’21. Obviously enough, we could hardly be aware of the results presented in this report when submitting our paper to NeurIPS’21.
>
> _``Also, although the paper makes a good point that the number of minimal sufficient  explanations is in general much smaller than the number of sufficient explanations,  it offers no evidence that the size of minimal sufficient explanations is significantly  smaller than the size of sufficient explanations.’’_
>
> The contribution of the paper does not reduce to computing minimal sufficient  explanations. The possibility to compute all contrastive explanations and to  generate explanatory features in polynomial time is interesting since such features somehow circumscribe the set of all sufficient reasons. Please see the last  paragraph before the conclusion.

---

> > ### Comment · Reviewer_DBdD · 2021-08-28
> > **Clarification.**
> >
> > The authors misunderstood the comment about proposition 5. The point was that the proof presented in the paper focuses on a concrete encoding of the problem, and that is unnecessary, as there are simple direct proofs. The review does not state that a concrete paper should be cited. The review states that the proof is unnecessarily complicated and mentions an example reference where a simpler proof can be found.

---

> > > ### Comment · Area_Chair_e6So · 2021-09-09
> > > **Authors response**
> > >
> > > Dear reviewer,
> > > Would you be so kind and post this comment to the reviewer?
> > >
> > > Thanks
> > > The AC

---

> > > ### Comment · Area_Chair_e6So · 2021-09-09
> > > **Response to clarification**
> > >
> > > Dear authors,
> > > Since there is no full agreement on this paper, your answer to the clarification above might be important.
> > >
> > > Thanks for your time
> > > The AC

---

> > > > ### Author Response · Authors · 2021-09-11
> > > > **Response to clarification**
> > > >
> > > >
> > > > Thanks for the clarification. Indeed, we misunderstood your comments, sorry for that.

---

### Official Review · Reviewer_7f2X · 2021-07-19

**Rating:** 7
**Confidence:** 4

**Summary:**

This paper considers the explainability of decision trees.  By conventional wisdom, decision trees are considered interpretable, although this paper explores this notion much more deeply.  Namely, they show that "good" explanations (in some specific sense) are not necessarily tractable with decision trees, and that explaining the behavior of a decision can still be a non-trivial problem.  The authors further provide another novel way of assigning weight to the importance of a feature in a classifier's decision.


**Limitations And Societal Impact:**

addressed

**Main Review:**

I generally have a favorable opinion on this paper, since it highlights how explaining decision trees is not a solved problem per se, even though almost all people dismiss the explanation of decision trees as uninteresting up front (see, e.g., funding agencies but also including me).

If expressed purely in terms of logic, then the given results are probably not too surprising.  When one considers the results in the context of explaining decision trees, then they become more interesting.  For example, some of the (negative) results emphasize to me that one should not limit oneself to subsets of the instances being explained (i.e., not just terms as used in prime-implicant explanations and Anchor explanations).


**Time Spent Reviewing:**

2

---

> ### Author Response · Authors · 2021-08-10
> **Response to the review by by Reviewer 7f2X**
>
> Thanks for your comments. Yes, you are perfectly right: our point was to show that explaining decision trees is not a problem that is fully solved, and to take a step further in the direction of explaining decisions made using decision trees.

---

### Decision · Program_Chairs · 2021-09-27

**Decision:**

Reject

**Comment:**

This is a borderline paper.
While decision tree is a fundamental classifier in machine learning, the paper addresses a more philosophical problem using tools from logic that may be not so relevant for most of the audience.
Another issue is the novelty: The authors confirmed that they missed the fact that Proposition 4 was already proved, and I suspect other results are also known. A much simpler proof for Proposition 6 is available online. While it was published after NeurIPS deadline, it is another indicator that some of the results in the paper are known, or at least can be explained in a more trivial way.